# Management Strategies in the Comprehensive Rehabilitation of the Historic Centers of Quito and Havana

**Juan Carlos Martínez Serra** [1],* [ID] **and Enrique Fernández-Vivancos González** [2],* [ID]

1 Facultad de Arquitecturay Urbanismo, Universidad UTE, Quito 51060, Ecuador
2 Department of Architectural Projects, Universitat Politècnica de València UPV, 46022 Valencia, Spain
* Correspondence: juanc.martinez@ute.edu.ec (J.C.M.S.); estudio@fernandez-vivancos.com (E.F.-V.G.)

**Abstract:** Historical centers are structural elements in contemporary cities which preserve identity and collective memory. Despite being lubricants of social cohesion, intense processes of urban growth, fragmentation, and degradation put these city centers at great risk. Thus, they have been considered priority spaces in public renewal policies affected by inaccurate interventions which must contend with changing and complex realities in the Latin American and Caribbean contexts. This article approaches the main management strategies used in the comprehensive urban rehabilitation of historical centers through critical and comparative analysis of the historic centers of Quito and Old Havana, which are two UNESCO World Heritage Sites. The study ultimately aims to determine the main successes and failures of the management strategies used and proposes measures to support decision-making processes, optimizing the type of urban intervention employed.

**Keywords:** historical centers; Historic Center of Quito; Historic Center of Havana; management strategies; integral urban rehabilitation

## 1. Introduction

Most Latin American and Caribbean cities have undergone a marked transformation of their physical structure, which is also reflected in the cultural values of their "historic centers" or "old towns". Both of these terms refer to old identifiable points of origin or central spaces in the urban structure which are linked to the collective memory that has been built based on the use of its citizens [1,2].

Commonly located in central areas of cities, these centers generally concentrate a high population density and are well endowed with supporting infrastructure, services, and urban public spaces with great capacity for attraction [3]. This is manifested through its centralizing role in modern metropolises which greatly influence the city's development factors. This leads to the creation of perpetual urban districts which steer periods of great social, cultural, and economic prosperity [2,4,5].

The current socioeconomic, patrimonial, and urban problems of Latin American cities are closely associated with its historic centers, which are in constant transformation due to their response to new and emerging needs. This is seen in the degradation and deterioration of the built environment, which does not have adequate conservation of infrastructures and heritage buildings. Moreover, this is also seen in the influence of technologies and the transformation of public spaces, the loss of cultural values, the growth of the population index (in some cases), and the gentrification. In other contexts, depopulation, high housing deficits, inequalities and exclusions, residential underuse, mobility difficulties, and the incompatibility of its urban fabric for automobile use fundamentally blind its centers to follow laws of the market economy [2,6–10].

Disasters and natural hazards, such as the earthquakes that occurred in Mexico City in 1985 and in the city of Quito in 1987, along with intense rains and rising sea levels, are perennial problems that have allowed for diversity in proposed solutions and responses on the conservation of historic centers. These diverse solutions constantly consider urban

rehabilitation actions implemented by local governments in search of sustainable alternatives, which are supported by value enhancement processes. Ultimately, these efforts aim to the rescue of identity, traditions, and the safeguarding of built heritage [11,12].

Hence, historic centers become subject to constant processes of urban rehabilitation to generate economic and cultural benefits from tourism and become symbols of novel urban rehabilitation strategies which encourage the existence of social, economic, and cultural groups. Meanwhile, politicians which ensure the continued existence of said groups become an active source for economic and sociocultural development [11,13,14].

Along with the rehabilitation efforts developed and implemented in many historic city centers in Latin America and the Caribbean, substantial changes in the physical–spatial structure have also manifested in the spaces. These efforts even tend to clash with current and future elements necessary for the conservation and rescue of cultural values [15,16].

These same city centers also face the mismanagement on both the planning and execution of conservation efforts, thus becoming museums or tourist centers instead of continuously being functional city districts. Because of its inability to house new residents, citizens are often displaced into "fortress"-type housing complexes which offset the mixture of what is originally a compact city. This dissolution of the connection between city and citizenship also contributes to extensive and speculative urbanization [4,8,11,17].

Notably, the regulation in urban rehabilitation actions in these areas also impose a conservationist model that contributes to the segregation and fragmentation of its originally integrated functions [8,14]. The resulting speculation and appropriations following the conservation efforts of protected landscapes affect cities and its historic centers, creating problems that are often difficult to solve at its core [18].

Historical centers feature a structuring element due to their centrality. Imposing intervention in the centers implies effects on the rest of the city. Hence, these centers represent a field of interest for both public and private institutions, especially those adjacent to the execution of comprehensive urban rehabilitation and/or recovery programs in different countries and localities [2,19,20].

Current approaches to management strategies for comprehensive urban rehabilitation are insufficient: there remains the clear and dire need to deepen the employed analysis of the execution of the developed plans and strategies for historic centers in the Latin American context [1,15,19,21].

The weight that historical centers carry in the urban structure of Latin American cities deems the deeper explorations both interesting and necessary for the planning, management, and commercialization of related goods and services in these locales [2].

The study explores the current problems and the methodical characterization experienced by the cities of Quito and Old Havana during the recovery of their historic centers. It compares and evaluates the management strategies of the comprehensive urban rehabilitation implemented to determine the main successes and failures of the strategies used and subsequently proposes improvement measures based on this.

The historic centers of Quito and Havana were chosen because of their similarity at the time they were considered historical heritage and because of their emergence and evolution over time, following the rules of the Spanish colonial era for cities founded during the fifteenth century. to the XVII. These cities are also represented and recognized cases for Latin America and the Caribbean for their level of conservation and protection of heritage, they have extensive urban planning instruments and carry out numerous urban rehabilitation processes with negative or positive impacts both locally and internationally with similar scales in population and number of inhabitants.

The structure aims to explore two specific and salient aspects in the experiences of Quito and Old Havana. First, it looks into the conceptual position of (or in) comprehensive urban rehabilitation. Next, it focuses on the management strategies used in the recovery efforts of these two cities.

The methodology used herein followed earlier studies on management strategies employed by Ramírez et al. (2020) and Muñoz (2008) [21,22] and obtained its results and

conclusions from the following: (1) the common aspects of the different management strategies implemented for the rehabilitation and recovery of the two cases studied; and (2) the successes and failures of these management plans.

*1.1. The Theoretical Framework*

The Athens charter of 1931 and the Venice charter of 1964 outlined further suggestions on proper architectural and urban conservation. The charter forwarded actions proposed for the restoration of the monuments that guaranteed the continuity of the architectural work and of the urban or rural site with particular civic and historical importance to a population. These documents enshrined the idea of heritage going beyond the monument and the subsequent need for its recue and conservation for the enjoyment of future generations. This pushed for the preservation of traditional city centers linked to specific urban and special plans for historic centers in its entirety [23].

After the creation of ICOMOS in 1965, the conservation and revitalization of areas of interest in historic centers garnered greater regard. Thus, the Quito regulations of 1967 established the particularities of the Latin American historic city and the "enhancement" on the rescue and reaffirmation of the city's cultural attributes through the conscious participation of the different actors [12,23].

In the 1970s, UNESCO conducted several international meetings in Europe and Latin America to formalize strategic agreements for ancient cities and traditional towns. This coincided with the European Charter for Architectural Heritage and urban planning, land use planning, and respect for the urban fabric through the Amsterdam Declaration [12,23].

Thus, the inclusion of the Historic Center of Quito in the World Heritage List (which takes place after the Colloquium of Quito in 1977) establishes the comprehensive conservation policies of historic centers. These years also show that an important evolution took place in the economic approach on the close relationship between heritage and society—where heritage was proposed as a source of an economic revenue. This resulted in the harmonious balance of societies and their value such as education integrated with goals of conservation and social justice [12,23].

These documents and historical moments established the understanding of urban rehabilitation, and they have favored and made more flexible the modes of action. This makes it necessary to reiterate the idea of the historic center as an extension of the monument, which generated a difference between the areas with the highest heritage value in the central urban areas and the periphery [12]. This brought with it the segregation of cities and prevented them from being studied as a single organism, which tried to solve the problems of the historic city only through the historical–architectural lens, foregoing values of both economic and social use. Currently, historic centers are understood to be rehabilitated within their own urban dynamics, and they are prompted to provide economic potentials for their respective cities [3,6,23].

Some of the potential contributions of historic centers include the concentration of the past and the present in the same locale, facilitating the connection of urbans spaces. Links are also established between the public and private sectors which are often hinged on obtaining solutions at the urban level, thus resulting in collaboration, new investments, and the propagation of the locale's cultural identity [12,23].

An important highlight is the development of the Bologna plan during the late 1960s which saw the rehabilitation of its historic center framed following and incorporating rescue initiatives from both public and private investments. This in turn allowed for the massive tourism boom in the region. A proposal was forwarded wherein involvement from the city inhabitants in its rehabilitation essentially rescued and conserved Bologna's city center through preservation, restoration, and rehabilitation policies [12,24].

This plan later on became the impetus for other European and Latin American cities to begin organizing and developing important urban plans to rescue their deteriorating historic centers and open an international debate centered around their conservation [12,24].

Another important document in the preservation of historic centers is the International Charter for the Conservation of Historic Cities and Historic Urban Areas (otherwise known as the Washington Charter of 1987) which was adopted at the ICOMOS General Assembly in Washington DC in October 1987. This charter complements the Venice Charter of 1964 and defines the values to preserve in historic centers. These include the historical character of the population or urban areas and all those material and spiritual elements that determine its image. The document explicitly states that these values must be preserved since any change could modify the authenticity of the interventions. Moreover, this also means the varying participation of the population in these changes and the rehabilitation and conservation of the historic centers [23].

Another document, the Lisbon Charter of 1995, pioneeringly clarifies the forms of intervention for urban rehabilitation as urban management strategies through multidisciplinary interventions of the physical conditions of the park built by its rehabilitation and installation of equipment, infrastructure, and public spaces, aimed at giving value to their social, economic, and functional potential of the resident populations, maintaining the identity and characteristics of the area of the city to which they refer [23].

The document also establishes the rehabilitation of key city center areas using techniques such as urban re-functionalization, which is aimed at revitalizing old city centers by promoting new activities that respond to current contexts. The revitalization often includes operations for uplifting the city's declining economic and social life, which is then applied to all areas of the city with or without marked identity and characteristics. Another technique is reaching urban renewal, where subsequent actions carried out imply the transformation of the existing morpho-typological structures in a degraded urban area [23].

Another important document is the Xi'an Declaration on the conservation of the environment of heritage structures, sites, and areas adopted in Xi'an, China by the 15th General Assembly of ICOMOS on 21 October 2005. The declaration considers international and professional interest (which was then) focused on the conservation of the environment, historical monuments, and on sites (akin to the Venice Charter of 1964) where national committees of ICOMOS held international meetings for the materialization of the document in correspondence with previous ones. These include the Nara Document on Authenticity (1994) and the conclusions and recommendations of the Hoi An Declaration on the Conservation of Historic Districts in Asia (2003), the Declaration on the Recovery of the Bam Cultural Heritage (2004), and the Seoul Declaration on Tourism in Asian Historic Cities and Areas (2005) [23].

Furthermore, there exists various references to the concept of environment used in UNESCO conventions and recommendations such as the Recommendation Relating to the Safeguarding of the Beauty and Character of Landscapes and Sites (1962), the Recommendation Relating to the Conservation of Natural Assets Cultural Heritage Sites Threatened by Public or Private Works (1968), the Recommendation on the Safeguarding and Contemporary Role of Historic Areas (1976), the Convention for the Safeguarding of the Intangible Cultural Heritage (2003), and, most especially, the Heritage Convention Cultural (1972) and its Guidelines, where the environment is considered as a value of authenticity and, as such, requires protection through the delimitation of zones of respect and the opportunity that these provide for international and interdisciplinary cooperation between ICOMOS, UNESCO, and other entities for the development of issues such as authenticity or conservation of historic urban landscapes as reflected in the Vienna Memorandum (2005) [23].

Hence, various international legal bases emphasize that the environment is fundamental, and its various contexts must be evaluated and studied as a basic prior action. This is especially salient when evaluating its hereditary impact. Hence, studies on the environment must be multidisciplinary and collaborative in nature.

Urban rehabilitation interventions remarkably handle different variables for their execution, which aims at sustainable balance, function, and form which concern the exchanges

between urban and architectural spaces. The relationship of the urban center on the entire city directly influences the environment, the technical infrastructures, and the services the local government unit is able to provide [1,8,15,25].

Hence, comprehensive rehabilitation refers to a broader form of intervention that encompasses the entire urban fabric, where processes of consolidation, conservation, rehabilitation, and new insertions of both buildings and infrastructures take place. This include urban spaces, which are contemplated in the long, medium, and short term [3,6,24,26,27].

Because urban rehabilitation processes lead to the improvement and recovery of the built stock of historic centers, these often involve the intervention of equipment, infrastructure, housing, and public spaces being capable of improving the locals' quality of life [3,6,14,24,27,28]. When changes and transformations seemingly do not cease, assisting the adaptation to the psychosocial and cultural needs and aspirations of citizens is essential. This requires constant readjustments of their homes and the entire context, thus influencing urban functions and forms and the movement of people both at the vehicular and pedestrian level [15,24,27]. Needless to say, urban rehabilitation must likewise be constantly interdisciplinary and collaborative in manner and approch.

As illustrated in Figure 1 below, seven important dimensions in urban rehabilitation are mentioned. In principle, the first five dimensions stress endogenous local development. The use of appropriate and advanced technologies, construction systems, and materials and the protection of natural resources and good management of environmental conditions are also key components of the last two dimensions [1,3,15,27,29,30].

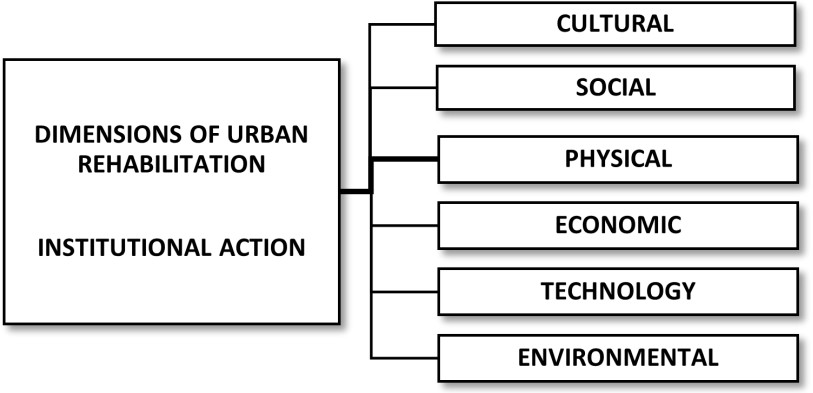

**Figure 1.** Scheme of strategic dimensions of urban rehabilitation.

For these interventions to materialize, it is necessary to carry out an in-depth assessment considering the previous dimensions and the experiences of other historical contexts, which is then coupled by mixed actions where different forms are combined; ranging from the conservation of the buildings to the rehabilitation, the renewal of public spaces, strips and areas, and economic renewal or regeneration where the preservation of heritage and its cultural values plays a key role. Urban improvement that is therefore undertaken through rehabilitation can be carried out on an "individual" scale or by "zones or areas" through different actions that partially regenerate neighborhoods or degraded areas, with coverage being essential for urban planning [1,3,8,16,27].

Figure 2 below shows the interventions that must be carried out on various scales along with their total summations until the appropriate combination is found. In this way, the structure can be better understood to rehabilitate through a detailed study of the different dimensions and variables and obtain design strategies according to the context [8,19,26].

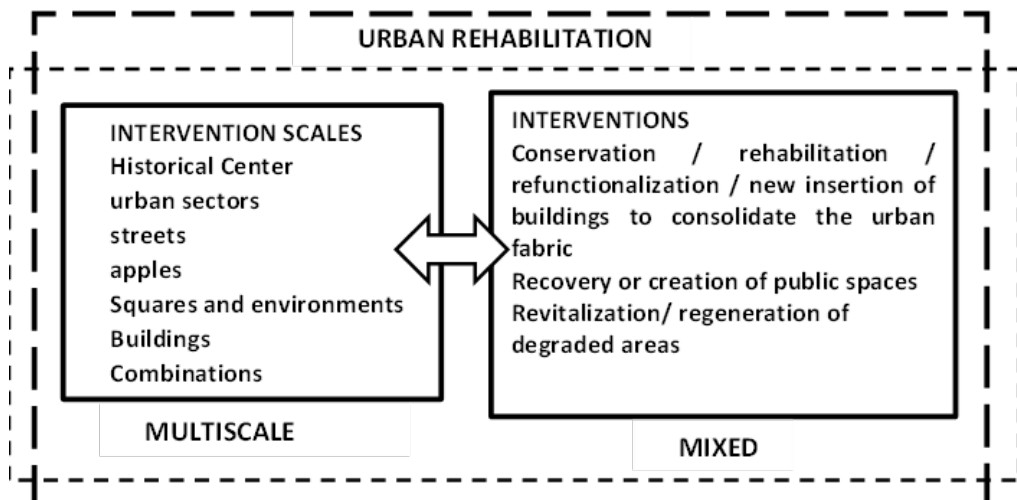

**Figure 2.** Scheme of strategic dimensions of urban rehabilitation.

Historic centers concentrate the largest amount of heritage, architectural, and environmental values of the city. Hence, the recovery of these not only implies physical and building rehabilitation and certain social and economic functions, it also includes the introduction of new functions and activities compatible with the current resources and demands of each context. To ensure that they are living areas and in correspondence with the needs of the population, an integrated local development based on sustainability, commitment, and social participation essentially must be formulated and executed [3,15,24,31].

In the article "The management of the integral development of the historical centers in Latin America", its author Patricia Alomá establishes a starting point of the manifestos on the conservation of the Historical Centers in Latin America: specifically, it stems from 1967 in Ecuador. When these regulations were issued in Quito, they were presented as pioneering and contemporary aspects of urban management in historic centers. For the first time, the guidelines were established for governments of the entire region to exert effort on a multinational level to reach these objectives [32].

The Quito document leads to a very interesting conclusion: the expression "enhancement" is questioned and enters an ethical conflict, since they relate expression to "commercialization" (which will be later be further explored vis a vis the "revaluation" of tourism) because historical monuments cannot be rehabilitated only voluntarily because of the need for enough sustainable resources. This leads to understanding that the contradiction is not between economy and culture but that of management control. The problem also lies not in the model being used but in the ethical positioning of the problem at hand [20].

In 1972, the United Nations Organization for Culture, Science and Education (UNESCO) draws up the Convention on the Protection of the World Cultural and Natural Heritage. From then on, countries worldwide were able to propose urban or natural sites which were of international interest and/or World Heritage Sites. This opened the opportunity to launch the appropriate programs and thus address the challenge that arises in each city [23].

This is where management plays a very important role, which is reflected in the use of the existing funds, the recovery of spaces for new functions according to the new way of life and in the challenge to the capacity of the manager who takes sides in terms of the role of history in architecture and the context in which comprehensive rehabilitation proposals are inserted [8,14,24,30,33].

Urban rehabilitation implies management through a set of actions that makes it possible to execute proposals contemplated in the plans, which are then translated into regulatory and financial instruments that then control the investment process, the creation of management, the use of endogenous resources, and the cooperation of actors and citizen participation, among others. These principles are implicit in the intervention models of

numerous countries, being references to consider when addressing this issue for the central areas, since these have successfully put rehabilitation processes into practice [3,8,30,33].

Among the historic centers in Latin America and the Caribbean where management plans are framed socioeconomically, heritage and urban values have been implemented for their comprehensive rehabilitation. The Master Plan for the Historic Center of Lima (1999) has also been highlighted, where the recovery of its downtown and surrounding historic neighborhoods constituted a priority for the execution of projects focused on housing, mixed uses, and commercial activities. From the beginning, it was committed to achieving the repopulation of the Historic Center through the reoccupation of unoccupied spaces in addition to establishing new administrative simplification procedures that improve operational levels and urban activities. It included, in addition to the recovery of monumental buildings, the elimination of slums and the construction of new homes for families living in precarious conditions as well as the development of soil improvement programs and archaeological investigations [34].

Another important example is the Partial Plan for the Urban Development of the Historic Center of Mexico City, which established since 2000 various management strategies to recover and strengthen the housing function as well as the promotion and consolidation of various economic activities that intended to support small businesses and businesses that are compatible with residential use. The current process for managing activities continues through the Comprehensive Management Plan for the Historic Center of Mexico City, which promotes urban and economic revitalization for the recovery of centrality, the growth of popular commerce, the improvement of public spaces, housing buildings, and heritage through adequate information and the protection of movable and immovable property [35].

*1.2. Management Strategies for the Recovery of Historic Centers*

Strategies are defined as decisions needed to achieve a specific objective or purpose. Understandably, these are not lasting— they must be reviewed and/or evaluated to develop new strategies [36]. The definition is then transposed into the urban theme, where two different actions are needed: on the one hand, to model or work on the provision of urban management instruments. On the other, it allows determining the times and the resource management processes. By combining both actions, a response is given to specific problems in an urban reality.

Accordingly, these strategies contribute to the comprehensive recovery in historic centers. Hence, all of them must be considered to respond to the complex problem of the deterioration of historical monuments. The following strategies, experiences, and objectives can therefore be defined Muñoz, 2008 [22]:

- Political–institutional strategy: instruments that allow directing urban recovery processes from state powers or institutions;
- Planning strategy: following urban analysis and strategic planning are programs developed which allow organizing a territory and projecting it into a possible and desired future scenario;
- Land management strategy: actions aimed at facilitating and speeding up the development of comprehensive recovery and urban renewal projects.
- Economic–financial strategy: directives aimed at knowing the various forms and possibilities of financing for the recovery of urban centers.
- Participation strategy: interactions and interventions of social, economic, financial, and political agents in the recovery processes of urban centers.
- Promotion and marketing strategy: policies which promote projects and programs for the rehabilitation of deteriorated buildings, the urban renewal of buildings in a state of physical and functional obsolescence, and the recovery of public space anchored by the value of sustainability.

## 2. Materials and Methods

In this study, the qualitative approach was used for the comparative analysis of the cases to obtain the common aspects of the different management strategies implemented for the rehabilitation and recovery of the two cases studied and the successes and failures of these management plans.

The methodology is structured in three stages as shown in Table 1 to determine the results of the investigation. In stage 1, the current Latin American problems and the historical evolution of the treated topic were theoretically investigated, focusing mainly on the historic centers of Quito and Havana and in this way defining the management strategies used for the comprehensive rehabilitation of these historic centers. For this, methods and techniques have used that range from bibliographic analysis and logical history to documentary analysis.

In stage 2, the representative historical centers in Latin America and the Caribbean were defined, and the characterization of these territories was outlined for their better understanding through analysis and synthesis and the observation of reality. At this stage, through the comparative analysis of the cases and the statistical processing and previous studies on management strategies used by Ramírez et al. (2020) and Muñoz (2008), the main management strategies were implemented in urban planning instruments for the comprehensive rehabilitation of the historic centers of Quito and Old Havana.

Finally, in stage 3, the main successes and failures are defined through the analysis and synthesis of the case studies shown in the comparative table of results.

**Table 1.** Methodological structure of the research.

| Stages | Research Design | Methods and Techniques |
|---|---|---|
| STAGE 1. THEORETICAL FOUNDATION. | Statement of the problem and research purpose. Definition of terms and concepts. Historical evolution of the subject of study. Selection of information sources. | Bibliographic analysis Historical–logical analysis Analysis and synthesis Documentary analysis |
| STAGE TWO. CHARACTERIZATION. | Definition of management strategies for comprehensive rehabilitation. Criteria for determining hits and misses. Definition and characterization of the study cases. Identification of the main management strategies implemented in the urban planning instruments for the comprehensive rehabilitation of the historic centers of Quito and Old Havana. Identification of the comparison parameters. | Comparative analysis of cases Analysis–Synthesis Observation of reality Statistical processing |
| STAGE 3. RESULTS. | Comparative table. Definition of the main successes and failures. CONCLUSIONS | Analysis–Synthesis Statistical processing |

## 3. Results

In Latin America and the Caribbean, after being declared UNESCO World Heritage Sites in the late 1970s and early 1980s, two representative rehabilitation experiences began in the historic centers of the cities of Quito and Havana.

*3.1. Historic Center of Quito (HCQ)*

Quito is the capital of Ecuador and is considered one of the most representative historical centers of Latin America. It has an urban area of 376 hectares and a population of approximately 40,000 inhabitants [37].

Of its 5000 buildings, approximately 130 are considered monumental and have high cultural values. It consists of a central core and a peripheral area with a road and parcel structure defined by a quadrangular grid that corresponds to the checkerboard pattern of the Indies, which then grew and adapted to the topography of the area [37].

However, far from being considered as one more part of the urban fabric of the city, its special historical, cultural, and geographical characteristics place it as a benchmark for the identity of the Ecuadorian people and heritage of a strategic nature for the economic development of the country (thus, it was declared by UNESCO as the first historical center Cultural Heritage of Humanity in 1978) [11,37].

Despite the great progress that has been made in recent years, the HCQ still has serious structural problems. Risk factors such as the poverty of certain social layers, the poor state of conservation of certain heritage buildings, the degradation of public space, the lack of equipment, and poor road accessibility are some of the symptoms that require decisive and comprehensive intervention [11,37].

During the end of the 1980s, the municipality of Quito simultaneously and comprehensively addressed the problem of rehabilitation of the central area of the city which then involved the private sector. This led to the establishment of the Spatial Structure Plan in 1993 [37].

After the incorporation of the HCQ in the UNESCO World Heritage List and the earthquake of 1987, awareness of the preservation of its heritage was realized and subsequently led to the creation of the Quito Salvage Fund (FONSAL). As the institution in charge of preserving HCQ, it created the Master Plan for the Conservation of the Historic Center in 1994, which was based on the Plan for the Metropolitan District of Quito and the Master Plan for the Comprehensive Rehabilitation of Historic Areas, which was dedicated to the restoration of monuments and buildings of high cultural value as well as the improvement of public spaces and infrastructure. Simultaneously, the Administration of the Central Zone was created, which was dedicated to meeting social demands, administering, regulating, and managing the execution of programs and projects [11,37].

Then, in 2003, the Special Plan was created with a comprehensive strategic vision of development until 2010 of rehabilitation of the HCQ, which presented proposals for the change of land use, public spaces, housing, and investments in the improvement of the image urban. In 2012, through the approval of Ordinance 0236, the regulation and control instruments were deepened and the promotion of tourist activities for the HCQ began [11].

Over time, the management model of the Historic Center of Quito was developed: today, there is already the Metropolitan Institute of Heritage, which has continuously received international recognition for the scope of the interventions and its integrity [37].

Notable city monuments include historic churches, mostly representative of the colonial baroque, as well as their residences and palaces which have been preserved as part of a rehabilitation process, in which the protection of intangible heritage is the main objective [38]. Other actions include the comprehensive rehabilitation of the former Cumandá bus terminal, which is now converted into a meeting place, for recreation, for sports and the well-being of citizens, the recovery of La Ronda Street and the 24 de Mayo boulevard. As can be seen in Figure 3, Plaza San Agustín, inaugurated in 2016, is located at the back of the heritage building Convento San Agustín. These projects have generated new green areas and public spaces for the use of citizens and the valuation of the urban memory of the place.

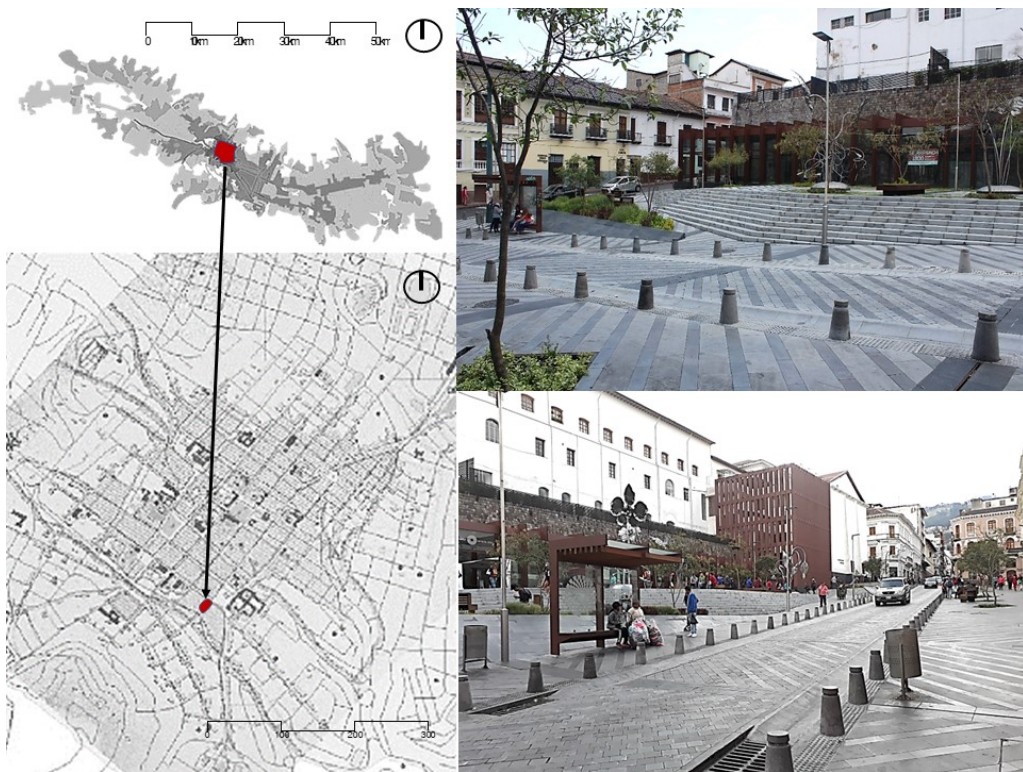

**Figure 3.** San Agustin Square of HCQ.

Notably, there was the renovation in 2019 of García Moreno Street, also known as Las Siete Cruces street, which has been made pedestrian to turn it into the Paseo de las Siete Cruces, in a section of the historic center that contains some of the main monuments of the city, such as the Carondelet Government Palace, the Church of the Society of Jesus and the Metropolitan Cathedral of Quito.

### 3.2. Old Havana Historic Center (OHHC)

The Historic Center of Old Havana, with an area of 214 hectares, is part of the municipality of the same name located in the Capital of Cuba with a population of approximately 70,000 inhabitants. This means 66% reside in the municipality. It is also considered one of the best-preserved historic centers in Latin America. Undoubtedly, it represents a monumental area with great cultural values present in the buildings that compose it [11,19,39].

Since the founding of the Office of the City Historian in 1938, where the first ideas for the conservation of the OHHC were born, the 1955 granting of the condition of "Zone of High Significance for Tourism" through agreement No. 2951, as well as the emergence of other institutions such as the National Center for Conservation, Restoration and Museology (CENCREM), led to the execution of conservation actions, thus gaining momentum after the inclusion of the OHHC in the UNESCO World Heritage List in 1982 [40].

Until the 1990s however, the appearance of the Master Plan for the Comprehensive Revitalization of Old Havana in 1994 within the structure of the Office of the Historian began the transformation of Old Havana with the support of other countries and was then declared a highly significant area for tourism [31].

Notably, there is a special "status" for the Historic Center at the government level, which is reinforced at the institutional level in Decree-Law no. 143, where the Council of State defines the Historic Center of Havana as a "priority conservation zone" [40].

At first, work was completed with greater intensity on the recovery of the area of what was "La Habana Intramuros": its monuments, streets, and squares, and on the complex social problems with different conflicts, which was fundamentally due to the high population index of the central areas of Havana. With the support of the Cuban

government, the institutions involved in the management and execution of the Master Plan managed to recover more than 33% of the Historic Center of Old Havana in just 10 years [3].

The plan also incorporated other areas of Havana which needed recovery interventions due to their cultural values, singularities, and location. This included the rehabilitation of the Chinatown and the section of the "Traditional Malecón" of the Centro Habana municipality that includes from El Paseo del Prado to Maceo Park, in addition to La Bahía de La Habana, other buildings and important streets of the city [3,15,41].

As part of the development of the plans for the urban rehabilitation of the OHHC, the Special Comprehensive Development Plan (PEDI) 2030 was also prepared, which includes, through the planning instrument, the necessary tools for land and urban planning and the comprehensive development of the Historic Center. The backbone of this plan is to consider culture as a fundamental element of development and people as the main subjects of rehabilitation to achieve comprehensive and sustainable development [39].

Likewise, the plan addresses socioeconomic problems, the preservation of heritage and cultural assets, and protects the environment, while achieving efficient and cultured exploitation of the potential part of the territories [39].

In the rehabilitation interventions of the OHHC, the recovery of its historic squares can be emphasized where Plaza Vieja stands out, as seen in Figure 4, which not only rescued a deteriorated square invaded by cars but also rehabilitated many buildings of high cultural value, which were converted into hotels, museums, shops, and a large number of services where the residence is intertwined with other functions as part of the important social process.

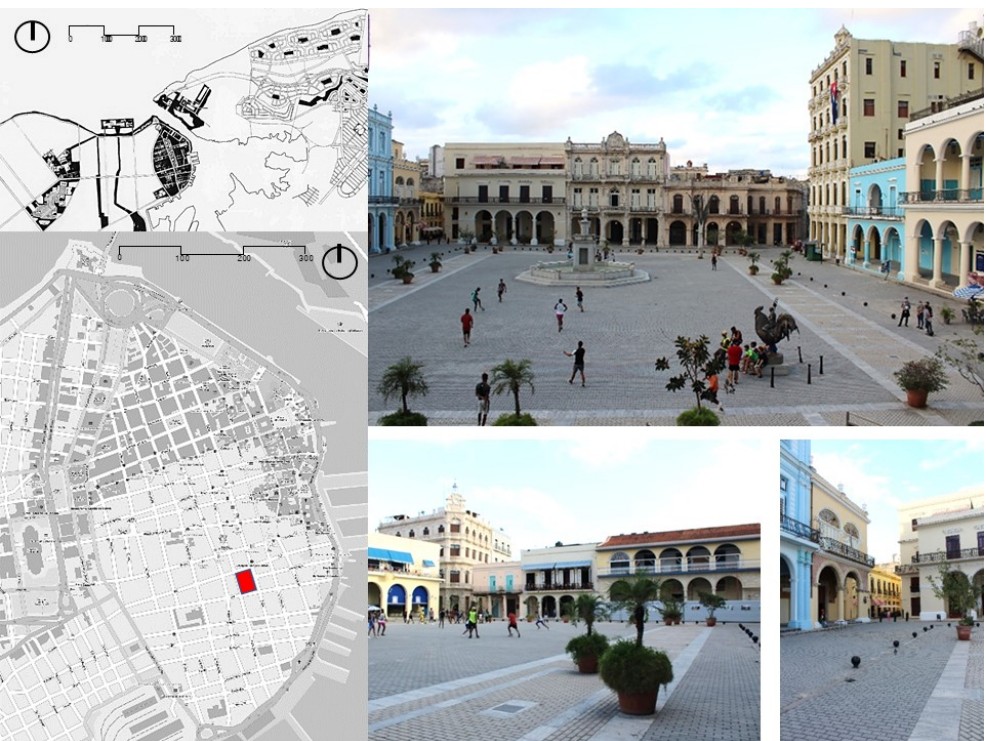

**Figure 4.** Old Square of OHHC.

After characterizing and identifying the urban planning instruments of the HCQ and OHHC, a comparative analysis was carried out and, as shown in Table 2 below, the different management strategies implemented for the comprehensive rehabilitation of the cases studied were defined, in terms of regarding the political–institutional parameters and their relationships with planning and economic resources, based on the contributions of Ramírez-Rosete et al. (2020) and Muñoz (2008) [21,22].

**Table 2.** Management strategies in the Historic Centers of Quito and Old Havana.

| Strategies | Historic Center of Quito (HCQ) | Historic Center of Old Havana (OHHC) |
|---|---|---|
| Political–institutional | Implementation of the Comprehensive Rehabilitation Program for the Historic Center of Quito, which developed the following projects: heritage conservation, social development, comprehensive redesign of traffic and transportation, decontamination and cleaning, citizen and property security, organization of the popular market, communication and "marketing" urban and traditional culture. | The Council of State of the Republic of Cuba approved a new law that redefines the functions of the Office of the City Historian, giving it the highest authority to promote the conservation and restoration of the Monumental Heritage and granting it legal personality, and the capacity to request, obtain and manage international aid. |
| Planning | Creation of the Company for the Development of the Historic Center of Quito, by the Municipality, which allowed restoring the heritage importance of the Historic Center. Reactivation of commercial activities and services, which favored the accessibility of citizens to the public services of government agencies. | Creation of a strategic alliance between the Office of the City Historian and the Spanish Agency for International Cooperation to study, at various scales, the problems of the Historic Center and the fortifications linked to the OHHC. To establish the most convenient strategies that can be carried out for its recovery. |
| Economic | Creation of a mixed economy company with independent and autonomous execution capacity. Management projects associated with the private sector | Development of a local economy state-owned and in mixed national–foreign associations has made it possible to accentuate territorial autonomy where an important part of what is produced is reverted to the recovery of heritage. |

In Table 3 below, the management strategies are defined by factors that are also of great importance, such as land management, the participation of the resident population, and everything related to the promotion of the Historic Center in its tourist aspects, which then generated events and the active functionality of squares and monuments.

Historic centers are (and will be) a certain possibility of preserving and promoting memory, generating senses of identity by function, and belonging. Ultimately, they are also a way to keep the constructions and buildings that are most representative of that same identity alive.

**Table 3.** Management strategies in the Historic Centers of Quito and Old Havana.

| Strategies | Historic Center of Quito (HCQ) | Historic Center of Old Havana (OHHC) |
| --- | --- | --- |
| Land management | The approach of real estate management instruments, which respect the typological configuration of the buildings, adapting and updating the internal conditions of the buildings to the current and future needs of operation and construction technologies. | Program for the repair or creation of housing, inside and outside the municipality, according to economic imperatives. As well as emerging actions in those houses with serious structural problems, and the recovery of buildings of social interest, among others. |
| Participation | Local participation observatory instrument. At the end of the first phase of the project, the Inter-American Development Bank (IDB) prepares a report on the operational management developed by the Historical Center Development Company (ECH). Three aspects are highlighted: social, economic, and institutional. | Participatory action is one of the keys that characterize the heritage management model developed in Old Havana. The model of the Office of the Historian of Havana achieved the integration of the resident population in rehabilitation projects and tourist activity. Here, the public consultation "Opening space" and "for your neighborhood" have been applied. |
| Marketing | Promotion of the Historic Center of the city as a tourist attraction. Organization of national and international cultural events. | Revitalization of the tertiary function in the Historic Center, associated with different cultural, commercial, gastronomic, administrative, and recreational activities, together with the real estate sector and the development of tourism mainly. |

Based on this principle, government management capacity for the conservation and rehabilitation of historic centers is essential. Strategies must be directed at their relationship with historical memory and must be treated as an integrated project with integrated goals. As can be seen in Table 4 below, this brings us closer to the successes and failures of the different management strategies in the Historic Centers of Quito and Old Havana.

Comparing Tables 1 and 2 and duly summarized in Table 4 below, there exists various coincidences or common aspects that were favorable (successes) and those that need to be addressed (mistakes) regarding the management strategies of the historic centers of Quito and Old Havana.

**Table 4.** Main successes and failures in the management strategies of the Historic Centers of Quito and Old Havana.

| Successes/Mistakes | Historic Center of Quito (HCQ) | Old Havana Historic Center (OHHC) |
|---|---|---|
| Successes strategies of management | Due to the managerial nature of the company, positive results have been obtained in the recovery of the Historic Center. The conservation of important buildings broadly reflects the cultural values in the emergence and evolution of the city. The habitability conditions of the area have been improved, in terms of the recovery and revitalization of public space. | The Master Plan and the Office of the Historian are present in each of the actions, projects and activities carried out at the HC in Havana. The Special Comprehensive Development Plan (PEDI) 2030 covers the four most important issues to promote the progress of a Historic Center: social, cultural, economic, institutional and environmental sustainability [39]. |
| Mistakes management strategies | Weaknesses have been identified in the operational management of the company, mainly in aspects related to economic control and management, which are listed below: (a) There is no concise procedure for prioritizing investments. (b) Proper project accounting is not kept. (c) The transfer of completed works by the company to the municipality is slow. (d) Commercial activities are not fully optimized. | Old Havana is beginning to suffer an unintentional process of gentrification, in which not only the Master Plan is involved but also the large multinationals dedicated to tourism, and even some local groups, which is generating a disengagement and a displacement of the population to other districts of the city [40]. In recent times, resource management has focused mainly on large tourism works and public spaces, with large areas persisting where the advanced deterioration of buildings can be seen. |

## 4. Discussion

It is important to highlight that in the historic centers of Quito and Havana, a comprehensive rehabilitation has been proposed where the political will accompanies the process and the authorities in charge coordinate the planning and the legal and technical management to develop the plans that consolidate the model. This is evident through the companies and institutions in charge of preparing and executing urban rehabilitation plans, which are aspects that are also highlighted in other Latin American centers such as the historic centers of Lima and Mexico City [13,14].

In the case studies, the complex institutional interaction for the elaboration of public policies and decision making that protect the historical and human value of their historical centers is manifested. However, the policies are directed fundamentally at real estate projects that promote the existence of hotels, restaurants, and activities oriented toward the tourism and services sector, taking advantage of the fact that the city center continues to be the most important destination for interurban traffic. Other historic centers, such as Lima, established new administrative simplification procedures in their initial plans to improve operational management levels [1,15,21].

Despite the fact that there is a clear trend for the recovery of HCQ and OHHC from their previous state of deterioration faced since the 20th century and beyond, there are still weaknesses in the management carried out by the companies and institutions in

charge where the difficulties and problems they continue to create in the processes of urban rehabilitation are evident in the deterioration of the housing stock, informal commerce, gentrification, depopulation and the quality of life of the resident population [1,15,19,21].

Undoubtedly, a lot of work has been completed in the recovery and conservation of the cultural values present in heritage buildings and public spaces, as well as in the preservation of history and social development. However, much remains to be completed with respect to the quality of life of the inhabitants where the growth of services dedicated to tourism is evident, which contrasts with inadequate housing and depopulation. This consideration is a significant point that is also manifested in several centers in Latin America and the Caribbean [15,16].

The study of heritage management strategies in these historic centers still shows some difficulties from the economic, political, and institutional spheres to conserve and preserve the cultural heritage that makes their historic centers a benchmark of identity and symbolism. This is evidenced by the fact that for the most part, the strategies have been aimed fundamentally at the tourism sector and at the change of land use from housing to commerce and services, which is manifested in the international promotion of these centers as tourist attractions, putting at risk the habitability of these places [13,14].

Although mixed economy companies have been created in these centers with the capacity for independent and autonomous execution and management of projects associated with the private sector and management plans, the endogenous potential for economic growth in these areas is not yet sufficiently exploited in the processes that are carried out with the available resources. This is in contrast to other historic centers such as Mexico City, where the promotion and consolidation of various economic activities present in their management plans support small businesses and businesses that are compatible with residential use [1,15,21].

The management instruments used in both cases have respected, above all, the typological configuration of heritage buildings and in many cases changing their use, in addition to the creation of programs for the repair and creation of new homes and buildings of social interest. However, to these changes and new adaptations, the current needs of the resident population inside and outside these historic centers are not fully met, when it comes to the increasingly growing development of the current market of cities in the world [2,3].

On the other hand, although participation is one of the keys to guaranteeing the management model in the historic centers of Quito and Havana, it can be said that there is still, to a large extent, a disconnection between the participation of the population and the rehabilitation processes. For this reason, more attention should be paid to citizen participation that really responds to the needs of the population and not as a way to fulfill initiatives, which leads to the need to find a different way of managing the rehabilitation processes that guarantee the enjoyment and permanence of the resident population [3,8,24,30,32,33].

Finally, although the urban rehabilitation of the HCQ and OHHC coincides and is focused on saving cultural values and social welfare with the incorporation of new facilities, services, and public spaces, it is essential to continue searching for alternative solutions for the improvement and necessary changes which facilitate urban transformation. This means the communication and the incorporation of the resident population to the different cultural and commercial activities to improve the quality of life of its inhabitants.

## 5. Conclusions

Among the main successes in heritage management strategies in the Historic Centers analyzed in this article is the motivation to return the use value to their historic buildings and the renovation of their public spaces as fundamental elements.

Undoubtedly, the interventions of the physical structure of these historic centers, with the use of new technologies and construction systems, have been a constant in conservation for 30 years (1990/2020), being a reference for the conservation of their heritage, increasing technical advances in urban planning in urban rehabilitation processes.

However, despite the fact that both cases started from a similar problem, there are still unresolved problems such as the issue of housing that influences the quality of life of its inhabitants as a result of the weaknesses in the management carried out by companies and institutions responsible for urban rehabilitation processes.

The policies aimed with greater interest at real estate projects, especially in the OHHC, promote the tourism sector as a result of management strategies that still do not resolve some difficulties in the economic, political, and institutional spheres.

The current needs of the resident population in these historic centers are evident, where there is still a great deal of decoupling between the participation of the population and the processes of urban rehabilitation, and the endogenous potential for the economic growth of these areas, still in progress, is not being exploited. These processes are carried out with the available resources.

This research contributes to strengthening the conceptual and methodological discussion of the current stream of literature, focusing on the intervention policies that are being applied in Latin American cities. Accordingly, it encourages continuing to apply management strategies that help protect the cultural values of historic centers, such as the cases of Quito and Havana, but it also delves into other aspects that have been applied in other Latin American and world centers to better understand and develop new forms of comprehensive rehabilitation for historic centers.

**Author Contributions:** Conceptualization, J.C.M.S. and E.F.-V.G.; methodology, J.C.M.S.; software, J.C.M.S.; formal analysis, J.C.M.S. and E.F.-V.G.; investigation, J.C.M.S.; resources, J.C.M.S.; data curation, J.C.M.S. and E.F.-V.G.; writing—original draft, J.C.M.S.; writing—review & editing, J.C.M.S. and E.F.-V.G.; visualization, E.F.-V.G. All authors have read and agreed to the published version of the manuscript.

**Funding:** This research received no external funding.

**Data Availability Statement:** Not applicable.

**Conflicts of Interest:** The authors declare no conflict of interest.

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
