# Peer review of "Management Strategies in the Comprehensive Rehabilitation of the Historic Centers of Quito and Havana"

_urbansci, doi:10.3390/urbansci7010004_

Round 1

Reviewer 1 Report

Your paper addresses an interesting topic, however some major improvements are needed.

The methodology used herein followed earlier studies on management strategies 83 employed by Ramírez et al. (2020) and Muñoz (2008)

This is the basic methodology you applied but it isn't explained in detail.

an important evolution took 108 place in the economic approach on the close relationship between heritage and society— 109 where heritage was proposed as a source of an economic revenue.

The connection between heritage preservation/restoration an its economic value is a very much studied issue on which you present no exmples similar to your study or relative references.

I find the discussion section very limited and mainly regarding wishful thinking about future planning. It should be elaborate and focused on the results also in comparison to other relative studies in the field.

The same should be, in turn, done in the conclusions section.

Author Response

Dear Reviewer

I send you the corrections made to the article based on your comments, which have been very opportune to improve the work.

Cheers

John C Martinez

Reviewer 2 Report

The paper is of interests especially because of the case study cities: both of them are of regional and international interests in the discussion of historic city preservation. However, concerning the structure and the quality of this paper, I have the following observations.

1.      In both figure 3 and figure 4,  the scale and the sign of compass are missing.

2.      Table 2 “Monumental Heritage and granting it legal personality”; repetitive names of the company..and put the elements in right order.

3.      Soil management should be changed to "land management"

4.      Discussion: in a standard academic work, the part of discussion is the platform where you really dialogue with other scholars and their works, it is the place where you position this paper within the academic debate. Therefore, you should make a series of “echoes” in concert with the works you cited in the introduction part. In the current version, the discussion part is really scarce. It is true that both cases are of interest in latin America context but there are many similar cities in other contexts (e.g. developing countries or developed countries ) that are experiencing the same development approaches in their historic cities,,so what is the particularity of this study? These are the points that should be clarified in this part.

Author Response

(The authors gave the same response as above.)

Round 2

Reviewer 2 Report

the discussion part has been improved, but in this version, there is no citation at all in the discussion part. The works of others and similar case studies should be adequately cited, like for the case of lima and mexico city.. this is a problem appeared in many other phrases and emphasis in this final part. the authors should cite more works to support their ideas and emphasis instead of self-evidencing.

Author Response

I agree with the observation, and we proceed to place the appointments according to the comments.
